# Characterization on Crack Initiation and Early Propagation Region of Nickel-Based Alloys in Very High Cycle Fatigue

**DOI:** 10.3390/ma15175806

**Published:** 2022-08-23

**Authors:** Zelin Chen, Zihao Dong, Chang Liu, Yajun Dai, Chao He

**Affiliations:** 1Failure Mechanics and Engineering Disaster Prevention Key Laboratory of Sichuan Province, Sichuan University, Chengdu 610105, China; 2MOE Key Laboratory of Deep Earth Science and Engineering, College of Architecture and Environment, Sichuan University, Chengdu 610065, China

**Keywords:** crack initiation, fine granular area, Inconel 625, rough area, stress intensity factor, very-high-cycle fatigue

## Abstract

As nickel-based alloys are more and more widely used in engineering fields for bearing cyclic loadings, it is necessary to study their very-high-cycle fatigue (VHCF) properties. In this paper, the fatigue properties of nickel-based alloy 625 were investigated using an ultrasonic fatigue test apparatus. The fracture microscopy shows that around the crack initiation site there are two characteristic zones, a rough area (RA) and a fine granular area (FGA). Inclusions caused the interior fatigue crack initiation, and the coalescence of neighboring micro cracks was strongly influenced by the local microstructure, resulting in the RA morphology. Subsequently, the contact and compressing of the crack surfaces contributed to the formation of the FGA. Finally, the stress intensity factors of the RA and FGA were quantitatively evaluated for further discussion of the crack initiation and propagation processes.

## 1. Introduction

Inconel 625 is a solid-solution-strengthening Ni-based superalloy with molybdenum (Mo) and niobium (Nb) as the main strengthening elements. Due to its high strength, excellent processability, weldability, and great resistance to high temperature and corrosion, it is widely used in aviation, chemical, petrochemical, and marine industries, especially in the manufacture of gas turbine engines, high-pressure turbine blades, nuclear power equipment, aerospace engines, and ship applications. With the rapid development of technology, the requirement for longer life and higher properties of these pieces of equipment and parts keeps increasing. In the process of service, alloy materials often need to experience very high cycles of up to 10^7^ cycles or more, and fatigue failure is one of the main causes of equipment failure. Therefore, it is necessary to study the very-high-cycle fatigue failure and crack initiation mechanism of nickel-based alloys. The process of fatigue failure beyond 10^7^ cycles for a metallic material subjected to cyclic load is called very-high-cycle fatigue (VHCF).

At present, there are many studies on the fatigue properties of Ni-based alloys; however, few of them focus on fatigue behavior and fracture mechanism under very-high-cycle service. It has been illustrated in previous research on the VHCF behavior of nickel-based alloys that fatigue strength decreases as fatigue life increases. The S-N curve shows a quadratic inflection, or bilinear form [1]. Moreover, the experimental results show that the temperature from room temperature to 1000 °C and the vibration frequency from 0–20 kHz have little effect on the very-high-cycle fatigue properties of Ni-based single-crystal/columnar-crystal superalloy [2]. Fatigue cracks initiate on the surface below 107 cycles and originate from internal discontinuous areas such as casting defeats or inclusions in the material in VHCF conditions [1,3]. Casting pores have a prominent contribution to the VHCF life of Ni-based alloys compared to chemical composition. When porosity is absent, the crack initiation site switches to other microstructural features like inclusions, which do not affect fatigue life too much [4]. In addition, there is a competitive mechanism between surface and internal crack initiation. When the crack initiates from the internal inclusions, the area of crack initiation and initial propagation shows the characteristics of a “fish eye” [5,6]. Initial crack is likely to appear at grain boundaries. A significant pile-up of dislocations in highly planar slip enhances intense impingement of slip bands at grain boundaries. As grain size increases, the grain boundary is weakened as the barrier for dislocation and short crack growth is easier. In addition, metallic carbides reduce the binding force, which promotes fatigue crack initiation. The crack propagation rate at an early stage is very slow because of the zigzag deflection from the ends of crack initiation [7]. The rough area (RA) section in the interior of the crack initiation area, which is proven to be a high-dislocation-activity area, is relatively rough, and its surface range grows larger as the number of cyclic loadings increases. A fine granular area (FGA) forms around the RA where cracks propagate slowly when cracks initiate from the internal inclusions, and it consumes most of the total fatigue life. After crack initiation around the inclusion, the repeated contact throughout a very large number of cycles leads to fine particles. [3,5,6,8,9]. The fish-eye section outside the FGA is much flatter, and is a smooth crack propagation zone. In the phase of crack propagation, the mixed propagation of intergranularity and cleavage is the main form [1,9,10].

The effect of microstructure on early crack propagation has been studied extensively, but the role of these effects on fracture morphology characterization has not been indicated. Previous studies on the causes of the formation of the FGA in Ni-based alloy are relatively scarce. The formation mechanism of FGA is very complex with the combination of very long life, local microstructure, external environment, and other factors. The effect of microstructure on the short crack growth behavior at an early stage has been studied. The short crack growth after initiation is likely to propagate along the slip system activated with high Schmid factors (SF) in the neighboring grains. The shielding effect of the γ’ variant results in a slower short crack growth rate [7]. However, few studies have focused on how the interior vacuum-like environment affects the crack propagation and the formation of the FGA. The internal vacuum-like environment has a significant impact on the formation of FGA in comparison with the case where no FGA is generated.

In this study, the fatigue crack initiation and early crack propagation of Inconel 625 under very-high-cycle fatigue were investigated with the help of a 20 kHz ultrasonic fatigue machine. This study aims to discuss the formation mechanisms of crack initiation and early propagation, as well as the formation of rough areas and the possible quantitative relationship between rough areas and fine granular areas.

## 2. Materials and Methods

In this investigation, Ni-based alloy Inconel 625 was studied, the main chemical composition of which is listed in Table 1. The superalloy was supplied by Shanghai Zhongnie Industrial Group Co. (Shanghai, China) as cast cylindrical rods 10 mm in diameter. The specimen was cut from the bar and machined to the dimensions of the fatigue test specimen. The specimen was polished in order to reduce the effect of surface defects on the fatigue test.

After the surface of the specimen was polished by mechanical polishing method, the sample was corroded in a mixture of 6 mL hydrochloric acid, 2 mL water, and 1 g CrO_3_ for 30 s, and then experienced ultrasonic cleaning with anhydrous ethanol solution for 1 min followed by hot air drying. Figure 1 illustrates the microscopic organization of the specimen observed by optical microscope (GX-53, Olympus Corporation, Ltd. Tokyo, Japan), which shows that the microstructure of the material presented equiaxed grains with an average size of 9.38 μm. The average grain size was calculated by dividing the number of intersections by the actual line length. Average grain size = actual length of the line/number of intersections.

Figure 2 shows the design of the test specimen. The central diameter of the sample was 3 mm, whose designed natural longitudinal frequency was 20 kHz, and the displacement–stress coefficient ratio was 26.9 MPa/μm. The uniaxial tension–compression fatigue test was carried out by the ultrasonic fatigue test system working at 20 kHz frequency. The axial strain was controlled by adjusting the loading displacement. The fully reversed (R = −1) strain control mode was utilized for all the fatigue tests. The principle of the test equipment is shown in Figure 3. In order to prevent the sample temperature from rising due to high-frequency vibration, the intermittent loading method was adopted during the test. Each loading time was 500 ms, and the gap was 200 ms. In addition, the compressed air-cooling method was used to control the temperature of the specimen in the room-temperature range. The step-loading test was performed before the fatigue test to obtain the appropriate fatigue load, during which the applied displacement gradually increased from 1 μm. If the specimen does not fracture at 10^7^ cycles, then 1 μm displacement is added and the test continues. At last, the specimen fractures with a displacement of 26 μm, which represents 699.4 MPa.

After the fatigue test, the fracture surfaces of all samples were observed by scanning electron microscope (SEM, JSM-6510, JEOL, Ltd., Tokyo, Japan). Three-dimensional crack characteristics were obtained by three-dimensional morphology scanning of specimens with 10^8^ cycles (IFM-G4, Alicona, Ltd. Graz, Austria).

## 3. Results

### 3.1. S-N Data of the Specimen

The S-N diagram was drawn according to the fatigue load and cycle number obtained from the test, as shown in Figure 4.

The ultrasonic fatigue data of the specimen showed stepped or bilinear forms, where there were two inflection points, located near 10^6^ and 10^8^ cycles, respectively. These data points could be divided into two groups, representing the high-cycle-fatigue (HCF) range with fatigue life less than 10^7^ cycles and the VHCF state with fatigue life more than 10^7^ cycles. It can be seen that the stress amplitude range of the left region was 690–710 MPa, where the fatigue cycle was less than 10^6^ cycles, in the HCF range. When the stress amplitude on the right side was in the range of 670–700 MPa, the fatigue cycle number was higher than 10^7^ cycles, which was in the range of the VHCF. The fatigue cycle data of these two states increased roughly linearly with the decrease in stress amplitude.

Combining the fracture morphology observed (Figure 5, Figure 6 and Figure 7) with the S-N data, it can be deduced that there was a certain relationship between the crack initiation position and the fatigue life; the crack initiated on the surface under the HCF range, whereas in the VHCF state, cracks often initiated at inclusions or defects in the interior or subsurface [1,3].

In the process of increasing fatigue load, the crack initiation location transfers from surface to interior, and there is a certain competition mechanism between internal and surface crack initiation. A similar appearance of the S-N curves in the VHCF has also been reported previously for high-strength steel, but it is rare for Ni-based alloy. It is believed that the surface oxide film inhibits surface crack initiation and promotes internal cracks under low fatigue load [11]. The stress concentration appeared on the surface under high stress played a major role leading to surface cracks at this time, and the internal stress concentration under low stress was the main factor causing internal cracks [12]. The load conditions of internal cracks are lower than those of surface cracks, and the internal cracks mainly appear when the surface initiation is suppressed. Internal crack initiation happens under low stress, and the rate of crack formation and propagation is slow. Fracture will lead to more cycles with internal crack initiation, resulting in a huge gap in the number of cycles between the two situations.

### 3.2. Fracture Morphology

With the increase in fatigue life, there are three modes of fracture morphology.

When the cycle number is less than 10^6^, the crack initiates mainly from the surface. The typical fracture morphology is shown in Figure 5. The stress amplitude was 685.95 MPa, and the cycle number was 2.359 × 10^5^ cycles. The fatigue crack initiated from the surface inclusion. The inclusions and defects on the surface in the HCF state caused stress concentration under loading. Under high-stress amplitude, the slip bands were squeezed and intersected at the inclusions or defects, resulting in surface cracks and multiple crack sources.

When the stress amplitude is lower than 688 Mpa and the cycle number is greater than 10^7^ cycles, the crack initiates from the internal surface or subsurface under the VHCF state. A fish-eye area appeared in the fracture morphology of the specimen, and the micro-graph is provided in Figure 6. The stress amplitude was 679.23 Mpa, and the fatigue life was 4.818 × 10^9^ cycles. When the crack initiates from the sub-surface inclusion with a rough area, it stops expansion as reaching the surface. In Figure 7, the stress amplitude was 685.95 Mpa, and the fatigue life was 1.302 × 10^9^ cycles. The fatigue crack initiated from the internal inclusions and formed typical fish-eye characteristics, with a rough area as well as a fine granular area. In addition, there were other internal inclusions existing near the crack initiation site with small sub-cracks forming around them.

Figure 8 shows an obvious fish-eye feature of a fracture morphology. The stress amplitude was 682.68 Mpa, and the fatigue life was 8.492 × 10^8^ cycles. The fish-eye area stopped expansion as it reached the surface of the sample, and the crack propagation area outside the fish-eye area presented a different pattern of expansion. A crack initiation site also formed near other inclusions in the fish-eye area, but no main crack formed. In the center of the fish-eye zone, a ring of the FGA was formed at the periphery of the rough area. There were multiple inclusions in the rough area. From the RA to the FGA, the average particle size decreased rapidly from 9.48 μm to 1.54 μm. When the crack propagated, the average particle size returned to 8.17 μm in crack propagation zone.

### 3.3. Three-Dimensional Morphology Scanning

The 3D-scanned morphological results of the fish-eye region of the specimen in Figure 8 are shown below.

From the three-dimensional morphology in Figure 9, it can be seen that the high and low undulation of the fish-eye area was not large, and the change was relatively gentle. There was an obvious border between the fish-eye area and the crack propagation area outside. From the three-dimensional morphology of the rough area in Figure 10, it can be seen that the rough area was composed of flat platforms of different heights, resulting in undulations changing rapidly and largely. In particular, the height difference between the two platforms in the rough area reached about 10 µm, which was similar to the grain size.

## 4. Discussion

### 4.1. Formation of RA

In a previous study, RA was rarely mentioned and its formation was hardly discussed. It was experimentally determined that the area of the rough zone increases with increasing fatigue cycles and decreasing fatigue load [9]. Vortex plastic flow forms a thin nanocrystalline layer and the nanograins rotate in each compression/tension cycle, and the loss of grain boundary strength makes the grain granulation (i.e., initial cracks), resulting in a rough section in the fish-eye region; however, the physical process remains to be verified [13].

In the fracture morphology of VHCF, the presence of rough areas at the crack initiation was observed, and there were always inclusions in the rough areas. The formation of the rough zone is related to the internal inclusions. The 3D scan data of the rough zone were obtained and a profile was made along the cut line to show the undulations in it. It can be seen from the Figure 11 that the average size of the depression and the bump was around 10.11 μm.

The undulation of the rough area was supposed to be the result of cracking along the grain boundaries. Under cyclic loading, initial cracks were generated around the internal inclusions, and the multi-cracks were not located at the same plane. Inclusions in the material caused decohesion between contiguous grains, which made the grain boundary strength decrease. In addition, their different elastic properties promoted an increase in stress concentration near the interface, making cracks easier to initiate. In the process of multi-crack coalescence, the cracks propagated along the grain boundary across the height, which corresponded to the shear fracture mode, leading to the formation of an undulating rough area.

### 4.2. Effect of Vacuum on the Formation of FGA

Observing the fracture morphologies of the VHCF specimens, some of the specimens formed a fine granular area around the rough area. For the specimens with cracks initiating from subsurface, only the rough area was formed, and the cracks directly entered the crack propagation area after the rough area touched the surface. The specimens with cracks initiating from internal inclusions showed a fine granular area, which was formed outside the rough zone in a vacuum-like environment.

There are models for the formation of the FGA in previous study. It was suggested that the fine grain layer is formed before the crack initiates. The micro-separation of the FGA initiates and converges to form a coin-type crack when it reaches a critical size, and the crack does not grow within the fine grain layer. In addition, some considered that the initiation crack front continuously forms the fine grain layer [14,15]. In the study of VHCF crack initiation in titanium alloys, it was noted that the FGA characteristics of the crack initiation are often formed in a vacuum environment, and it was concluded that the reciprocal contact of the initiating crack surface in a vacuum-like environment results in a “cold welding” process, which leads to grain refinement in the initiation zone, and the crack surface undergoes a “cold welding” process. However, the process is not experimentally based [16].

It is known that fatigue test results and the formation of the fine granular area are influenced by experimental methods, equipment, and environmental conditions. A fine granular area only appears when the crack initiates from internal inclusions. Compared to surface crack initiation, the main difference is that the initial environmental condition of FGA is vacuum-like. The vacuum-like conditions under which cracks initiate from the internal inclusions play an important role in the formation of the FGA.

A previous study showed that as for steel, fatigue life is longer in vacuum condition than in air and that the fracture surface is rougher in air [17]. In addition, it was found that an air environment can accelerate crack propagation but does not change the fracture mode [18,19,20,21]. Fracture morphologies vary greatly in air and vacuum condition and in a vacuum-loading environment, the fatigue crack propagation rate could be retarded significantly [22,23,24].

The internal vacuum-like environment retards the crack propagation, allowing an increase in fatigue life, which results in a numerous cyclic pressing between crack surfaces and thus the formation of rougher morphologies. Then FGA is formed and small cracks propagate slowly. Until the FGA expands to a certain area with its stress-intensity factor *ΔK* reaching a critical value, as is seen in Table 2 small cracks coalesce into the main propagating crack, thus forming a fish-eye zone. After the fish-eye zone reaches the surface, air enters the crack tip and promotes crack expansion. Cracks have different propagation rates and patterns between the fish-eye zone and the crack propagation zone due to the air-loading environment.

### 4.3. SIF and Crack Propagation Process

To further investigate the nature of fatigue crack initiation and extension behavior, the fracture was quantitatively analyzed using fracture mechanics theory, and the stress intensity factor (SIF) *ΔK* was calculated for the rough area of the fracture surface of the failed specimen. In this test, specimens with cracks initiating from the surface and the interior of the specimen were selected. The area of the RA where the cracks sprouted was projected onto a plane perpendicular to the load and measured. Using the square root of the area, the value of *ΔK* for surface cracks was calculated based on the expression of the maximum value of stress intensity factor *ΔK* for cracks initiating from surface (1) or internal (2) defects given by Murakami et al. [25].
(1)$$\Delta K=0.65\Delta \sigma \sqrt{\pi \sqrt{area}\;}$$
(2)$$\Delta K=0.5\Delta \sigma \sqrt{\pi \sqrt{area}\;}$$
where σ_0_ is the maximum stress amplitude for cyclic stress with R = −1, 0.65 is the calculated parameter for surface cracks, 0.5 is the calculated parameter for internal cracks, and “area” is the projected area. In the table, the cracks of specimens numbered 1–3 initiated from the subsurface, and the cracks of specimens numbered 4–6 initiated from the interior, with the presence of fine grain areas outside the rough areas. The projected area of the rough area in the SEM fracture photograph was measured, and the *ΔK* value of each specimen was obtained by substituting into the formula to make a stress magnitude–stress intensity factor plot. It is obvious in Table 3 that the RA projection area of the specimens with cracks originating from the interior was approximately three times that of the specimens with subsurface crack initiation, and it can be seen from Figure 12 that the *ΔK* values were very similar and that the average values for the subsurface and internal crack groups were 5.44 MPa·m^1/2^ and 5.66 MPa·m^1/2^, respectively. The crack initiation moved from the subsurface to the interior as the value of *ΔK* increased.

**Table 2 materials-15-05806-t002:** *ΔK* value of cracks initiating from surface and internal defects (FGA).

No.	4	5	6
Fatigue life	8.492 × 10^8^	1.302 × 10^9^	1.567 × 10^9^
Stress amplitude (MPa)	692.68	685.95	679.23
FGA area (µm ^2^)	15,689	14,794	16,179
*ΔK* (SIF) (MPa·m^1/2^)	6.87	6.70	6.79
Average	6.79		

The *ΔK* values of FGA when cracks initiated from surface and internal inclusions are shown in Table 3. The average *ΔK* value of FGA reached 6.79 MPa·m^1/2^, close to the threshold *ΔK* of Inconel 625 according to ASM, which is 7.1 MPa·m^1/2^, corresponding to a long fatigue crack growth rate of 10^−7^ mm/cycle [26,27]. This also partially supports the fact that crack propagation takes a long time to form FGA.

As discussed above, the crack propagation model is summarized in Figure 13. Under lower stresses, the stress concentration promotes initial cracks around the internal inclusions. Multiple cracks are not in the same platform, and during the process of coalescence cracks propagate along the grain boundaries, forming undulating a rough area. The initial fracture surface is in a vacuum-like environment, which retards the crack propagation. It enables the numerous pressing between fracture surfaces, making the particle size around the RA decrease. Meanwhile, small cracks are formed, resulting in a fine granular area. Then the number of small cracks increases and starts to coalesce with the main crack, and the crack propagation rate is very slow, at 10^−7^ mm/cycle. The *ΔK* value reaches a threshold value when the FGA reaches a certain area, and the small cracks start to integrate with the main crack and form a fish-eye zone. The fracture surface undulates more gently in vacuum-like environment and the crack propagates stably, which can be expressed by the Paris Law. When the crack reaches the surface, the air enters the interior, accelerating the crack propagation, and the crack propagation outside the fish-eye zone is faster and the fracture surface is rougher. Then crack propagation area reaches a certain area when *ΔK* is 40 MPa·m^1/2^; in the Paris regime, all the specimens demonstrated similar behavior.

This paper mainly discusses the formation of RA and FGA, and proposes the Inconel 625 crack propagation model. the effect of a vacuum-like environment on the formation of FGA is firstly discussed in Ni-based alloy. However, the study is limited due to lack of comparison between vacuum and air-loading fatigue tests. Further study is expected to focus on the effect of vacuum and air environments on microstructure in fatigue tests.

## 5. Conclusions

The VHCF behavior of Inconel 625 was experimentally investigated to reveal the effect of local microstructure on crack initiation and FGA formation. The main conclusions are as follows:In a very-high-cycle fatigue regime, fatigue crack initiate from the inclusions at the subsurface or interior of the specimen, and RA surrounding the inclusions can be characterized at the crack initiation site.The profile of the RA is sensitive to the grain size of the materials. The formation of the RA is a synergistic result of multi-crack initiation from the inclusions and the local microstructural effect.The fine granular area only existed in the case of interior crack initiation, which is related the numerous cyclic pressing processes in a vacuum environment.

## Figures and Tables

**Figure 1 materials-15-05806-f001:**
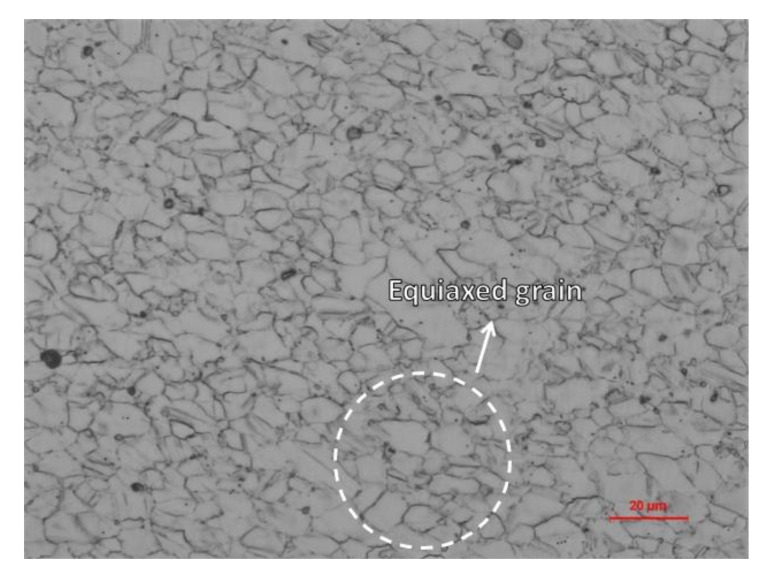
Optical micrograph of Inconel 625.

**Figure 2 materials-15-05806-f002:**
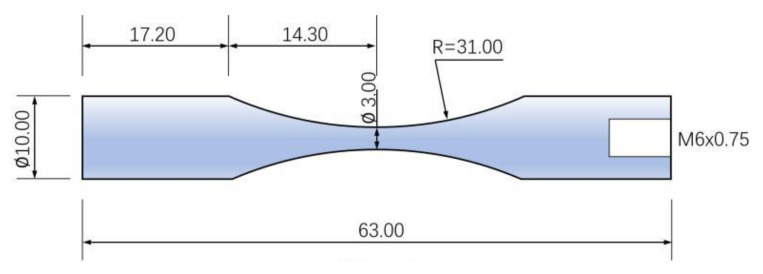
Geometry and dimensions of the test specimen (mm).

**Figure 3 materials-15-05806-f003:**
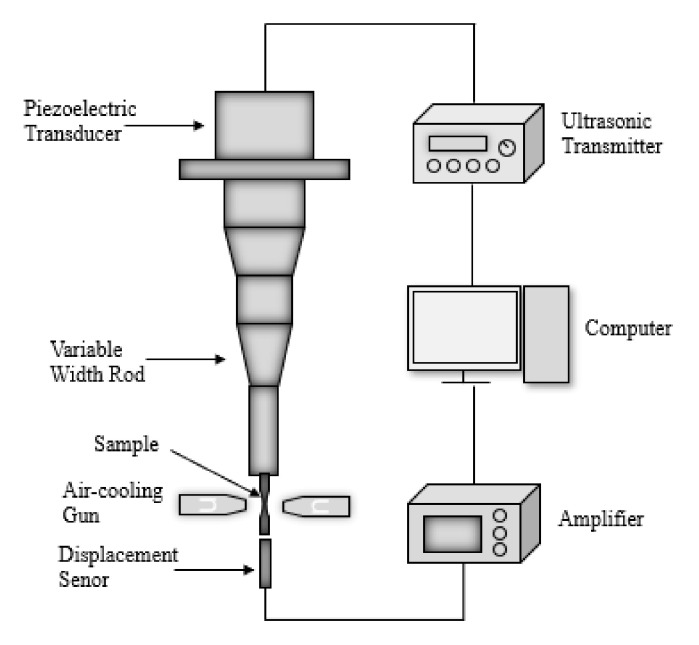
Principle diagram of the ultrasonic fatigue testing equipment. The fatigue specimen is fixed at one end of the displacement amplifier, and the mechanical vibrations generated by the piezoelectric transducer are increased to the required amplitude for fatigue loading through the displacement amplifier, causing the specimen to resonate at 20 kHz. The output displacement of the amplifier is measured by a dynamic displacement transducer.

**Figure 4 materials-15-05806-f004:**
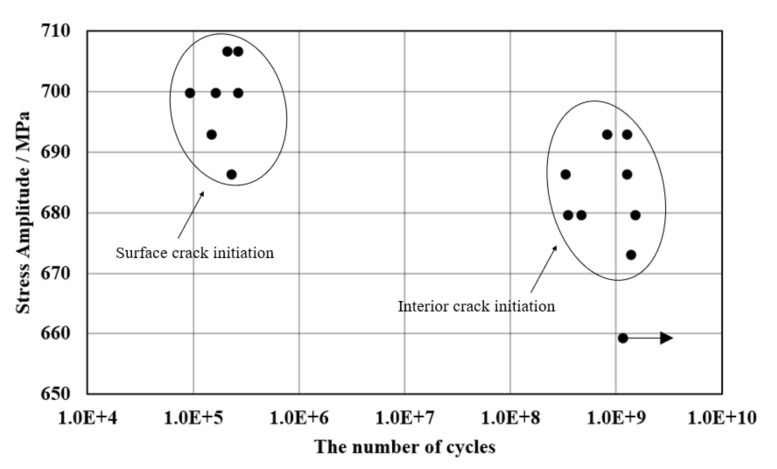
VHCF fatigue strength of Inconel 625.

**Figure 5 materials-15-05806-f005:**
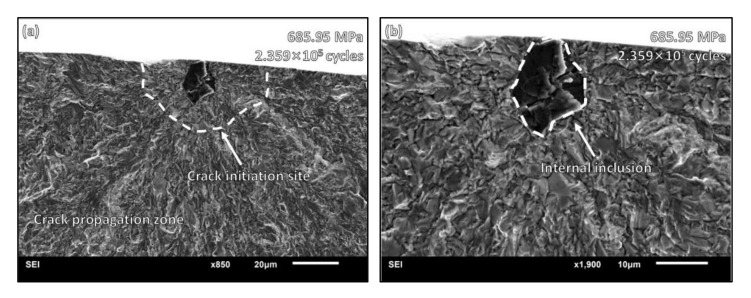
HCF fracture morphology: (**a**) crack initiation site on surface; (**b**) surface inclusion where initial cracks begin propagation.

**Figure 6 materials-15-05806-f006:**
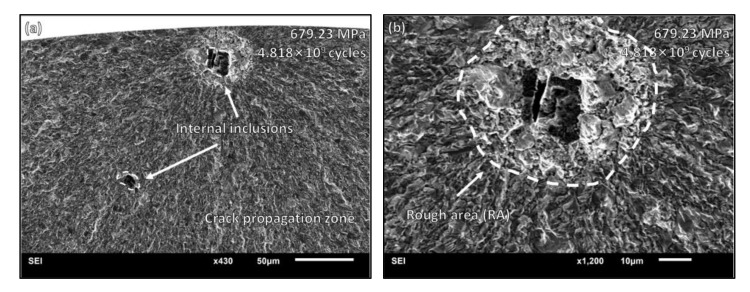
VHCF fracture morphology: (**a**) sub-surface inclusion distribution around the crack initiation site; (**b**) sub-surface rough area with relatively larger particle size and crack propagation zone outside the rough area.

**Figure 7 materials-15-05806-f007:**
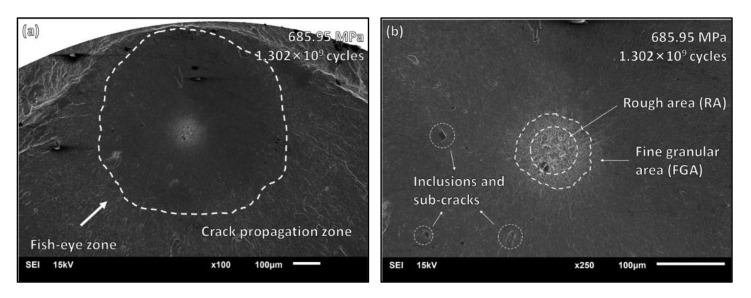
VHCF fracture morphology: (**a**) fish-eye zone from internal cracks, which stops propagation when reaching the surface of the specimen; (**b**) rough area and fine granular area. There are other inclusions around the crack initiation site where sub-cracks are formed.

**Figure 8 materials-15-05806-f008:**
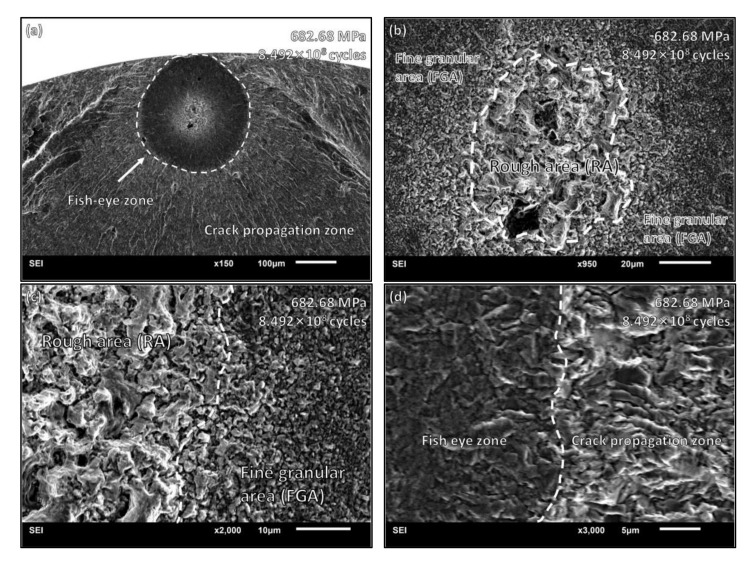
VHCF fracture morphology with FGA: (**a**) a typical fish-eye zone in VHCF state; (**b**) RA with internal inclusions and FGA around RA; (**c**) boundary of RA. Particle size decreases rapidly from RA to FGA; (**d**) boundary of fish-eye zone. Different crack patterns are shown in these two areas.

**Figure 9 materials-15-05806-f009:**
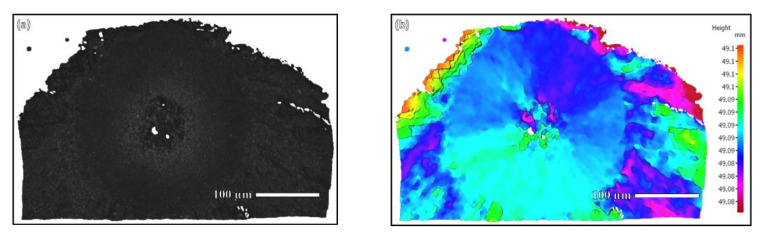
3D morphology scanning of fish-eye region: (**a**) scanning area of the fish-eye zone; (**b**) 3D contour of the fish-eye zone. The fish-eye zone is relatively flat compared to the outside crack propagation zone.

**Figure 10 materials-15-05806-f010:**
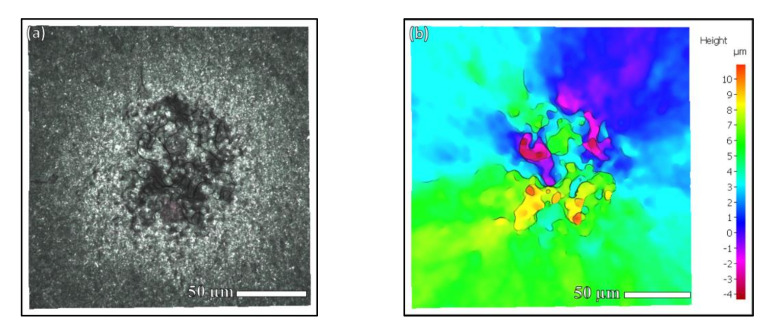
3D morphology scanning of RA: (**a**) scanning area of RA; (**b**) 3D contour of RA. There are several platforms with different heights in the RA resulting in large undulation.

**Figure 11 materials-15-05806-f011:**
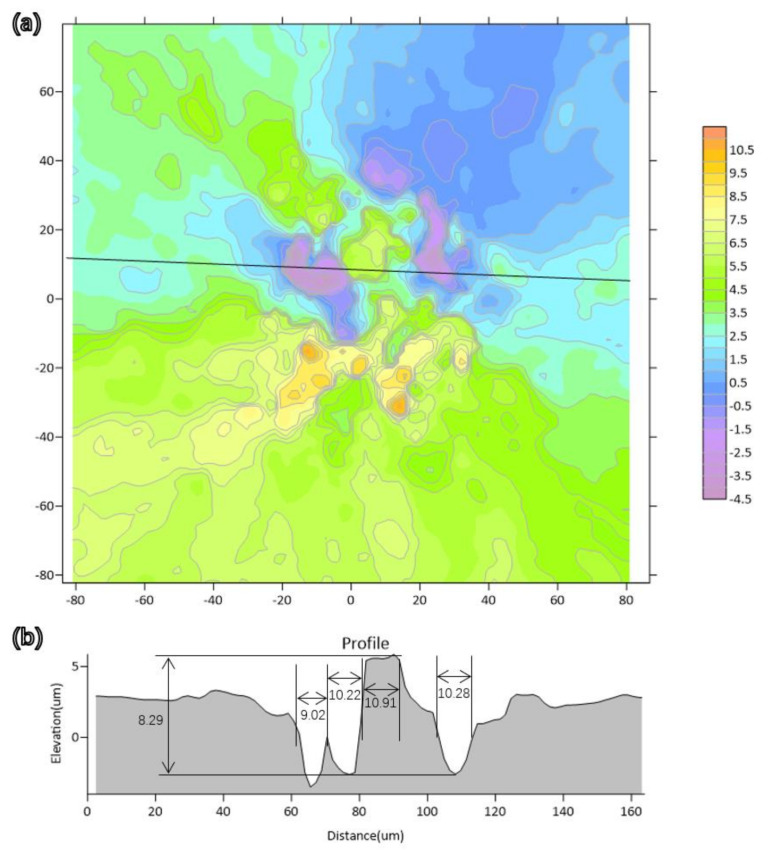
3D morphology scanning and profile of RA: (**a**) contour of RA, (**b**) profile of the cut line. There is a series of valleys and peaks in the RA and the average distance of a valley or peak is 10.11 μm.

**Figure 12 materials-15-05806-f012:**
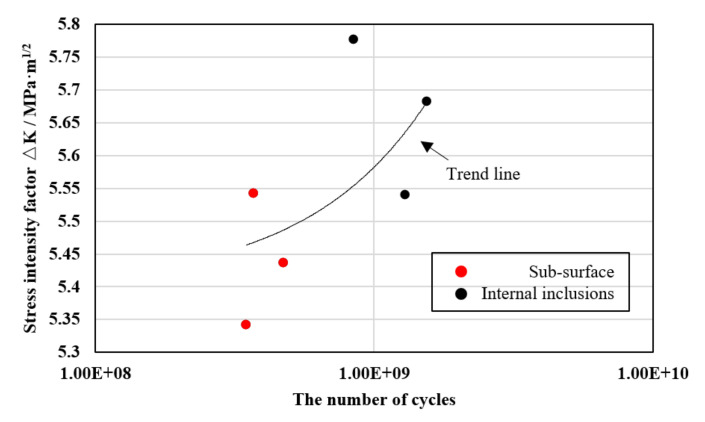
*ΔK* of specimen with surface and internal crack (RA).

**Figure 13 materials-15-05806-f013:**
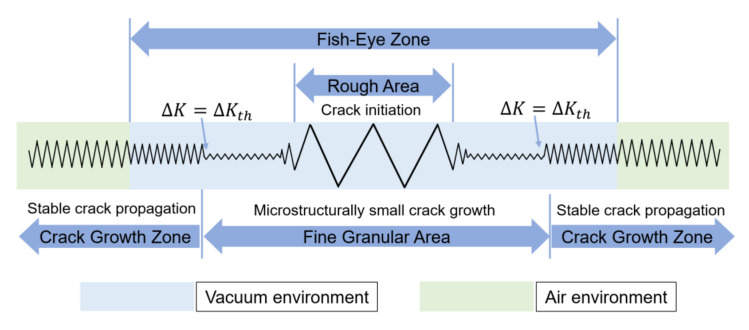
Inconel 625 crack propagation model (internal crack).

**Table 1 materials-15-05806-t001:** Chemical composition of Inconel 625 alloy.

Element	C	Cr	Fe	Mo	Nb	Mg	Al	Mn	Si	Ti	Ni
Weight (%)	0.053	21.32	0.11	8.58	3.73	0.01	0.18	0.04	0.09	0.16	residual

**Table 3 materials-15-05806-t003:** *ΔK* value of cracks initiating from surface and internal defects (RA).

No.	1	2	3	4	5	6
Fatigue life (cycles)	3.492 × 10^8^	3.734 × 10^8^	4.818 × 10^8^	8.492 × 10^8^	1.302 × 10^9^	1.567 × 10^9^
Stress amplitude (MPa)	685.95	679.23	679.23	692.68	685.95	679.23
Projected area (µm^2^)	2081	2508	2323	7830	6883	7923
*ΔK* (SIF) (MPa·m^1/2^)	5.34	5.54	5.43	5.77	5.54	5.68
Average	5.44			5.66		

## Data Availability

The data produced in this study are available from the authors upon reasonable requests.

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
