# Peer review of "Characterization on Crack Initiation and Early Propagation Region of Nickel-Based Alloys in Very High Cycle Fatigue"

_materials, 2022, doi:10.3390/ma15175806_

Round 1

Reviewer 1 Report

Please find the comments on each section:

1. Introduction: The reviewer feels that this section does not reflect the current state of art sufficiently. The knowledge gap and the novelty of the work proposed do not come from the introduction as it is based on 8 papers only papers. Please discuss the recent literature and clearly indicate the novelty of the presented research and what makes the authors' work valuable in comparison to other works already published. The current topic is well discussed thus the proper justification of the author's motivation for such work will be a crucial factor to consider their work for publication.

https://doi.org/10.1080/02670836.2022.2070336
https://doi.org/10.1007/s11661-018-4672-6

https://doi.org/10.1016/j.prostr.2020.01.080

2. Materials and methods: Please clarify how the average grain size was calculated.

3. Materials and methods: Please provide a description of how the ultrasonic fatigue testing equipment is actually working. In other words, describe its principles based on Figure 3.

4. Results: How the range of stress amplitude was determined?

5. Results: Please provide a standard stress-strain curve of the tested material.

6. Results: Lines 107-110: "Combining the fracture morphology observed with the S-N data, it can be deduced that there is a certain relationship between the crack initiation position and the fatigue life, the crack initiates on the surface under the HCF range, while in the VHCF state, cracks often initiate at inclusions or defects in the interior or subsurface" - please support with the proper reference

7. Results, Figure 4: There is a huge gap in the number of cycles to failure for the strain amplitudes of ~695 MPa and ~685 MPa. Please explain and discuss.

8. Discussion: The authors have performed their tests without a protective atmosphere or vacuum. What is the point of discussing the effect of vacuum on the formation of FGA then? Please clarify.

9. Discussion: The initial microstructure is one of the main factors that affect the fatigue life of the material. How the specimen for HFC was prepared then? Was it cut from the bar? How the homogeneity of the material in terms of internal inclusions was ensured? How the uniform grain size of the material was ensured?

General comments: The paper contains some interesting results presented in a good manner however, the scientific soundness and discussion are not at a satisfactory level. Firstly, please extend the abstract to highlight current knowledge on the topic and then clearly expose the knowledge gap. Secondly, please provide a proper discussion by a detailed comparison of results obtained with these found in the literature. Please support your fractographic observations with the proper references. The reviewer will reconsider this paper after major revision.

Reviewer 2 Report

The present study addresses the crack initiation process due to cyclic fatigue load in nickel alloys.

Here goes a few considerations:

The title and Abstract looks acceptable, as are the keywords, although in this last case I would recommend to place them by alphabetic order.

I recommend the authors to add in the body of the manuscript the meaning of VHCF also.

I suggest the authors to place a null hypothesis after the aim sentence.

How were the results of Table 1 obtained?

Regarding the sentence: “After the fatigue test, the fracture surfaces of all samples were observed by scanning electron microscope (SEM, JSM-6510, JEOL).”… may the authors be more clear regarding the model and city and country of origin?

In table 2 please add the measurement unit for “fatigue life”

The Discussion is generally fine. I just would like to suggest a point that I believe it might be interesting just to mention. The production and manufacture of the final goods may be also lead to possible issues that may affect the fatigue life. I think it worth to expose on interesting point mentioned in the study DOI: 10.1111/iej.13529 in which certain goods made of Nickel alloys were associated with early crack propagation marks even before use which may be related with the manufacturing process. So I believe it worth to highlight that the manufacturing process is highly relevant to attain a high fatigue life result in real world.

I recommend the authors to debate in the Discussion the study strength, limitations and further studies perspectives. A short paragraph would be welcome.

Please review the references in the list, I believe many of them are not according to the journal guidelines.  

Round 2

Reviewer 1 Report

The authors have corrected the manuscript according to suggestions thus I would recommend further processing and considering the paper for publication.

One comment to the authors: the stress amplitude for fatigue testing is usually determined based on the uniaxial tensile test results. The highest value of stress is then selected as ~80% of YP so the material will work in its elastic range. Based on the tensile curve provided by the authors, the significantly lower stress amplitude for fatigue testing should be considered as the current range exceed the yield point considerably.

Author Response

Thank you very much for your precious comments.

Reviewer 2 Report

Dear authors, i have no further comments. 

Author Response

Thank you very much for precious comments.